# Efficient palladium-catalyzed electrocarboxylation enables late-stage carbon isotope labelling

Gabriel M. F. Batista [1], Ruth Ebenbauer [1], Craig Day [1], Jonas Bergare[2], Karoline T. Neumann [1], Kathrin H. Hopmann [3], Charles S. Elmore [2], Alonso Rosas-Hernández [1] ✉ & Troels Skrydstrup [1] ✉

Carbon isotope labelling of bioactive molecules is essential for accessing the pharmacokinetic and pharmacodynamic properties of new drug entities. Aryl carboxylic acids represent an important class of structural motifs ubiquitous in pharmaceutically active molecules and are ideal targets for the installation of a radioactive tag employing isotopically labelled $CO_2$. However, direct isotope incorporation via the reported catalytic reductive carboxylation (CRC) of aryl electrophiles relies on excess $CO_2$, which is incompatible with carbon-14 isotope incorporation. Furthermore, the application of some CRC reactions for late-stage carboxylation is limited because of the low tolerance of molecular complexity by the catalysts. Herein, we report the development of a practical and affordable Pd-catalysed electrocarboxylation setup. This approach enables the use of near-stoichiometric $^{14}CO_2$ generated from the primary carbon-14 source $Ba^{14}CO_3$, facilitating late-stage and single-step carbon-14 labelling of pharmaceuticals and representative precursors. The proposed isotope-labelling protocol holds significant promise for immediate impact on drug development programmes.

The introduction of carbon isotopes ($^{13}C$ and $^{14}C$) into strategic positions of pharmaceutically active molecules is a crucial step for performing metabolic and pharmacokinetic studies (DMPK), the results of which ultimately determine their fate as potential drug candidates[1,2]. Similarly, in the search for new agrochemicals, their corresponding carbon isotopologues allow studies regarding metabolic profiling and the establishment of any possible human, soil, and groundwater contamination[3–5]. As such, it is imperative that efficient and accessible synthetic labelling technologies are at hand to facilitate late-stage installations of the isotope label into scaffolds of prospective drug entities. This provides rapid access to these essential isotopologues and avoids multiple manipulations of radioactive materials and waste for radiolabelling.

Aryl carboxylic acids and their derivatives represent an example of a highly abundant structural motif in numerous bioactive structures[6]. Classical approaches for accessing their isotopically labelled variants include the carboxylation of preformed but reactive organometallic reagents or nitrile substitutions followed by hydrolysis (Fig. 1a). Nevertheless, the harsh reaction conditions associated with such processes are characterised by poor functional group compatibility[7–9], thus limiting applications to the labelling of chemically less elaborate structures. A significant step forward was the development of transition metal-catalysed carboxylations, which substantially reduced previous functional group limitations and thereby increased the scope of these transformations[10,11]. A major challenge in isotope labelling, particularly for carbon-14 insertion, is

[1]Carbon Dioxide Activation Center (CADIAC), Novo Nordisk Foundation CO2 Research Center, Interdisciplinary Nanoscience Center, Department of Chemistry, Aarhus University, Gustav Wieds Vej 14, Aarhus C, Denmark. [2]Early Chemical Development, Pharmaceutical Sciences R&D AstraZeneca, Gothenburg, Sweden. [3]Department of Chemistry, UiT—The Arctic University of Norway, Tromsø, Norway. ✉e-mail: arosas@chem.au.dk; ts@chem.au.dk

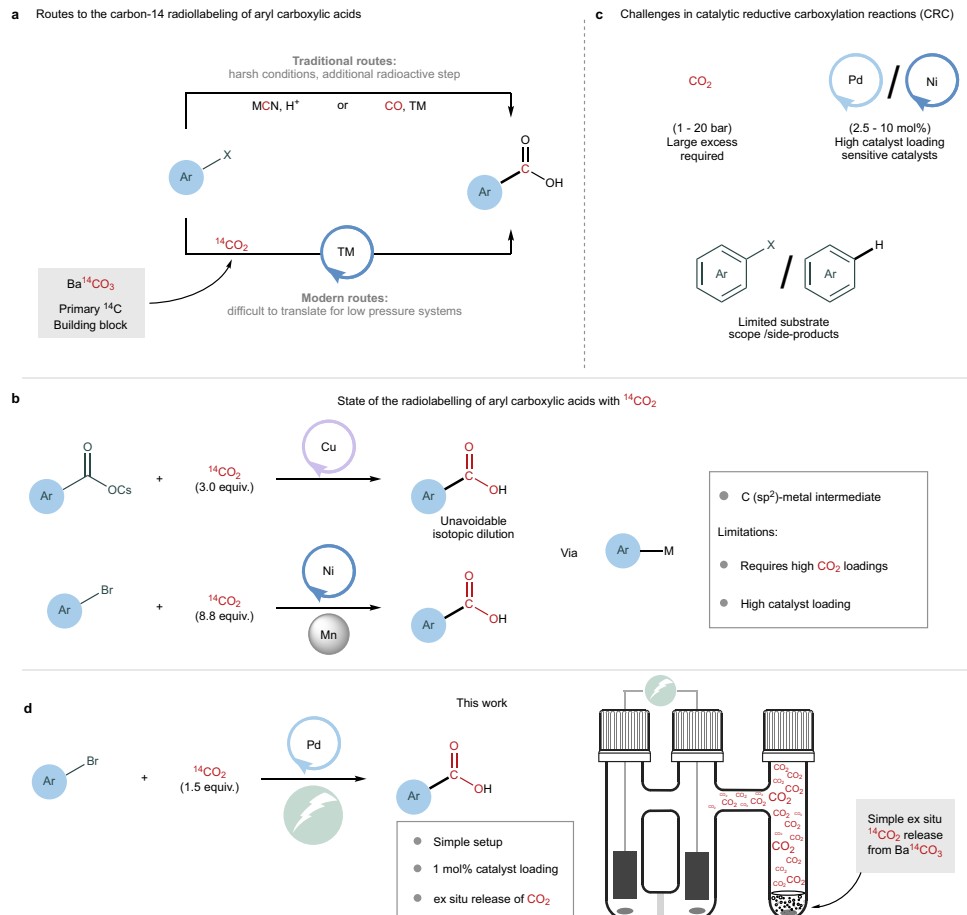

**Fig. 1 | Radioactive labelling of aryl carboxylic acids. a** Comparison of using $CO_2$, CO or MCN as carbon-14 sources for the radiolabelling of aryl carboxylic acids. **b** Top Modern dynamic CIE method limited to low-carbon isotope incorporations[12]. Bottom Modern strategy for radioactive CRC, limited to only one example and requiring high $CO_2$ loadings[29]. **c** Challenges in catalytic reductive carboxylation reactions (CRC). **d** Proposed method for the late-stage palladium-catalysed electrocarboxylation of aryl electrophiles (this work).

the need for efficient chemistry in the presence of only stoichiometric amounts of labelled $CO_2$. An example of a successful dynamic carbon isotope exchange was reported by Audisio and coworkers[12] for the decarboxylation–carboxylation of caesium benzoates catalysed by Cu (Fig. 1b). Although elegant and direct, full carbon isotope incorporation is challenging because of the equilibrium between labelled and unlabelled $CO_2$ under the reaction conditions. We and others have also demonstrated the usefulness of isotopically labelled carbon monoxide as a reagent in only stoichiometric or even substoichiometric quantities to access isotopologues of aryl carboxylic acids and esters[13–15]. Nevertheless, as $Ba^{14}CO_3$ represents the primary source of the carbon-14 isotope, an additional $CO_2$-to-CO reduction step is required to generate the corresponding labelled carbon monoxide.

The catalytic reductive carboxylation (CRC) reaction between organometallic species and aryl electrophiles constitutes an alternative route for directly installing a carboxylic group on an aromatic core. These efficient processes, which rely on transition metal-based catalysts including palladium[16–19], cobalt[20], copper[21], and nickel[22–28], nevertheless require a sacrificial reductant to generate a low-valent metal–aryl intermediate and form a new carbon–carbon bond through a $CO_2$ insertion step. Representative examples of such electron-donating reagents include stoichiometric metal and organic reductants (Mn, Zn, Organic Electron Donors (OEDs), etc.), or tertiary amines for reductive carboxylations applying visible light photoredox catalysis[16,19,26–28]. For example, Martin and coworkers reported the use of stoichiometric diethyl zinc with a palladium catalyst for the effective carboxylation of aryl bromides, where high

$CO_2$ pressures were required (up to 10 atm) to avoid the formation of the Negishi side-product[16]. Later, the use of photoredox catalysis in combination with palladium for CRC was disclosed by Martin, Iwasawa, and Jana[17–19]. To prevent parasitic hydrodehalogenation, several additives and 1 atmosphere of $CO_2$ were required[17–19]. Advances in the use of nickel catalysts for CRC were also successful, first with Tsuji reporting the use of metallic manganese as the reductant, followed by König demonstrating a photoredox setup[25,27]. Later, Martin and coworkers adopted Tsuji's CRC protocol for the radioactive labelling of a single aryl carboxylic acid, although with an excess of $^{14}CO_2$ (Fig. 1c)[29]. Altogether, it is uncertain whether these contemporary CRC reactions, when run with only (sub)stoichiometric amounts of labelled carbon dioxide, would prove sufficiently effective for accessing the desired labelled aryl carboxylic acids, rather than following alternative but unproductive pathways.

Reduction of the metal–aryl intermediate in an electrochemical setup represents an alternative scenario for the generation of the key low-valent species necessary for the carbon dioxide insertion step, with the advantage of avoiding the requirement of stoichiometric reactive reductants. In the early 1990s, Fauvarque, Jutand, and Torii reported the first successful electrochemical CRC using palladium- and nickel-based catalysts. Although seminal, the scope included primarily structurally non-elaborate aryl electrophiles, relying on atmospheric $CO_2$ pressures and high catalyst loadings (e.g., 7–10 mol% of $(PPh_3)_2PdCl_2$)[30–34]. More recently, Yu and coworkers. demonstrated electrochemical nickel-catalysed reductive carboxylation, but relying on high catalyst loading and excess $CO_2$,

significantly limiting this work's adaptability in radioactive labelling[23].

Herein, we present an innovative electrochemical approach for the late-stage carboxylation of aryl (pseudo)halides using [13]C- and [14]C-labelled carbon dioxide. Through systematic optimisation, we identified that Pd(BINAP)Cl₂ effectively facilitates the electrocarboxylation reaction under stoichiometric $CO_2$ concentrations, requiring only 1 mol% catalyst loading (Fig. 1d). This straightforward yet robust method leverages the ex situ release of $CO_2$ from $BaCO_3$, offering adaptability for the generation of both labelled and unlabelled $CO_2$. Notably, our approach avoids the stoichiometric use of highly reactive compounds as reductants. Furthermore, we showcased the practicality of this strategy for the late-stage incorporation of carbon isotopes into complex pharmaceutically relevant molecules in a single chemical step. With minimal investment in electrochemical equipment, our method provides a carbon labelling technology that is anticipated to find widespread use in pharmaceutical drug development.

## Results

### Initial considerations

Inspired by the pioneering work of Jutand, Fauvarque, and Torii[30–35], we set out to develop a general cross-electrophile coupling reaction in an electrochemical setup that provides efficient entry to aryl carboxylic acids with full carbon isotope incorporation (Fig. 2). The efficacy of this process hinges on two main factors. First, C–C bond formation through $CO_2$ insertion into the aryl metal bond should be possible at the targeted low $CO_2$ concentrations. Second, the aryl-metal intermediate formed must withstand the highly reductive potential required to drive the electrochemical process (Fig. 2a). In contemporary carboxylation methods, the aryl-metal intermediate in the catalytic cycle generated after the initial oxidative addition step can also participate in off-cycle reactions, such as the hydrodehalogenation reaction in the presence of protons or Negishi couplings in the presence of $Et_2Zn$[16,36]. These parasitic reactions have hampered efforts to adapt (sub)stoichiometric amounts of $CO_2$ for CRC reactions. It is essential to maintain a low proton concentration to minimise these undesired side reactions and exploit the advantages of electrochemistry in reducing the proposed aryl-Pd(II) intermediate and facilitating $CO_2$ migratory insertion. Protons can be present due to acidic motifs or from the oxidation of alkyl amines as sacrificial reductants, which, as for the case of the Pd-catalysed photoredox carboxylation of aryl bromides, requires the addition of two to three equivalents of caesium carbonate as a base[17–19,36]. Another approach for controlling the proton inventory in the cathodic chamber is to regulate

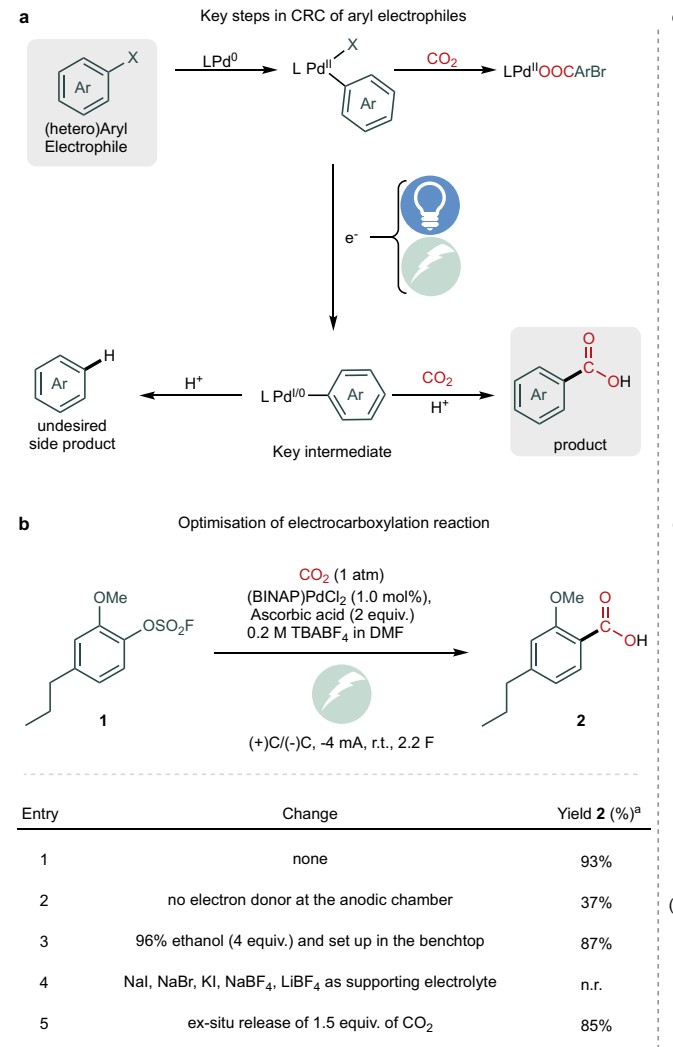

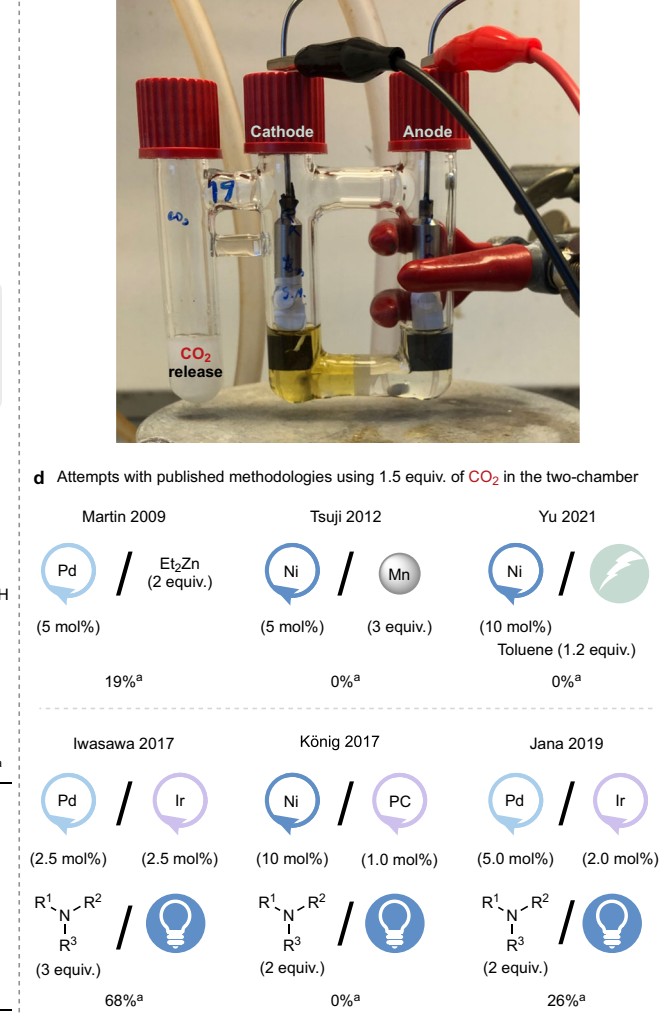

**Fig. 2 | Development of a palladium-catalysed electrocarboxylation method.** **a** Key steps involved in the CRC of aryl electrophiles. **b** Selected optimisation results. Unless otherwise noted, all reactions were performed in an H-cell with 0.4 mmol of aryl electrophile (1 equiv.) and 3 mL of DMF in each chamber (see SI Section 2). **c** The reaction setup. **d** Performing published CRC methods with 1.5 equivalents of $CO_2$ in the two-chamber (see SI Section 3)[16,17,19,23,25,27]. [a]Yields were obtained by [1]H NMR spectroscopy of a crude reaction mixture with an internal standard.

the proton flux between the cathodic and anodic chambers in a compartmentalised electrochemical cell, thereby avoiding the base additive. In addition, within electrochemical-mediated carboxylation reactions, the redox potential of the on-cycle intermediates is of utmost relevance. Using a strong electron-donating phosphine ligand favours the oxidative addition step; however, it makes the reduction potential of the Pd complex more negative. If catalyst reduction occurs at more negative potentials than the substrate, unfavourable substrate reduction can occur, highlighting the importance of the ligand used.

## Electrocarboxylation optimisation

With these considerations in mind, we optimised the Pd-catalysed electrocarboxylation of aryl electrophiles (Fig. 2b). Our efforts were directed to include the carboxylations of aryl bromides, because of their commercial accessibility, and phenol derivatives, because of the high natural abundance of such substrates. Aryl fluorosulfates were chosen as the pseudo-halide electrophilic partner because of their stability and ease of access from the corresponding phenols. Initial optimisation studies were conducted using the sterically hindered and electron-rich aryl fluorosulfonate **1** (see Supplementary Information Section 2). To our delight, the previously depicted limitations could be overcome by exploiting the commercially available catalyst (BINAP) $PdCl_2$ using an electrochemical approach (Fig. 2b). Optimum conditions were achieved using an H-cell with tetrabutylammonium tetrafluoroborate ($TBABF_4$) as the supporting electrolyte, carbon paper electrodes (2 cm$^2$), and DMF as the solvent. These conditions resulted in high selectivity and efficiency with only 1 mol% catalyst loading (Fig. 2b, for more detailed optimisation, see SI Section 2). The possibility of using a homemade galvanostat (ElectroWare) to apply a constant current of −4 mA further simplified the reaction setup[14]. After optimisation, carboxylic acid **2** was obtained in an 85% yield with only 1.5 equivalents of carbon dioxide, which was released in a three-chamber electro-glassware (Fig. 2c) using camphorsulfonic acid and barium carbonate (Fig. 2b, entry 5 and see SI Section 2). This was achieved without the use of metallic reductants and with the lowest reported catalyst loading for a palladium CRC reaction. Furthermore, only a 8% yield difference was detected when the $CO_2$ concentration was lowered from 1 atm to 1.5 equivalents (Fig. 2b, entries 1 and 5). The choice of barium carbonate as the $CO_2$ source provides convenient access to $^{12}$C-, $^{13}$C- and $^{14}$C carbon dioxide. Using an additive, including ascorbic acid, ethanol, or triethylamine, in the anodic chamber was essential for obtaining good yields of the carboxylic acid (see SI Section 2). In the presence of the additive, the potential at the cathode is not limited by the rate of the oxidation reaction since such additives should display a lower oxidation potential compared to the solvent, thus obtaining the product with higher yields.

To compare our results with previously reported methods, we performed a CRC of **1** with only 1.5 equivalents of $CO_2$ released in the two-chamber reactor, COware®[16,17,19,23,25,27]. The respective nickel-catalysed methodologies failed to provide the product with near-stoichiometric $CO_2$ (Fig. 2d and see SI Section 3)[23,27]. However, product formation was observed in the reported Pd-catalysed CRC reactions with substantially lower yields while employing higher catalyst loadings[16,17,19].

## Substrate scope

Having obtained suitable reaction conditions, we investigated the substrate scope for the Pd-catalysed electrocarboxylation reaction using aryl fluorosulfates and aryl bromides as electrophiles under atmospheric $CO_2$ pressures (Fig. 3, cmpds **3**–**10**). Electron-rich or hindered aryl bromide electrophiles give lower yields due to more demanding oxidative addition, whereas electron-rich or hindered aryl fluorosulfates provide the desired product in good yields. In the case of electron-poor electrophiles, aryl bromides react promptly, whereas

aryl fluorosulfates give lower yields, which depend on the reduction potential of the substrate. An electron-withdrawing motif decreases the reduction potential of aryl fluorosulfate, which can lead to substrate reduction as a side reaction. The side-product of this reduction is the corresponding phenol, following a reduction mechanism likely to be similar to that proposed by Jutand et al. for aryl triflates[31,34]. The optimised conditions also tolerated a variety of functional groups, including chloride and tosylate substituents, as for compounds **14** and **15**, demonstrating the high chemoselectivity of the reaction. Carboxylation at two positions under slightly modified conditions (−1 mA instead of −4 mA) was feasible, leading to bisphenol-A derivative **18** in 95% yield.

With good yields for the reaction using 1 atm of $CO_2$, we proceeded with the scope using 1.5 equivalents of $^{13}CO_2$ for stable isotopic labelling. Heterocycles are often used as a tool by medicinal chemists to optimise the vital characteristics of a drug candidate, such as solubility, lipophilicity, and hydrogen bonding capacity[37]. The reported method provided labelled and unlabelled heteroaryl carboxylic acids in good yields (Fig 3. cmpds **19**–**21**). Phenols are often found in nature, some of which have important biological activity; thus, we devised a late-stage derivatization of these natural compounds via the reaction of the phenol with sulphuryl fluoride, which leads to the corresponding aryl fluorosulfonate that can be applied in the electrocarboxylation reaction. With these, a library of natural compounds (derivatives) could be obtained (cmpds **2, 22, 23, 26**–**28**). Furthermore, the late-stage stable labelling of adapalene (**24**), an acne treatment pharmaceutical, and the antineoplastic drug bexarotene (**29**) was performed in 50% and 28% yields, respectively. Dynamic carbon isotopic exchange represents a rapid tool to access isotopically labelled active pharmaceutical ingredients (APIs). Nevertheless, this labelling technique generally does not provide full isotopic incorporation because of the dilution of the carbon-labelled $CO_2$ equilibrium with the released $^{12}CO_2$[12,38,39]. With the aim of developing a strategy for the full carbon isotope exchange of carboxylic acids, we envisioned that treating aryl carboxylic acids with the Cu-catalysed decarboxylative bromination developed by Macmillan and co-workers would provide an aryl bromide that can be further transformed to the labelled aryl carboxylic acid by applying the electrocatalytic reaction conditions[40]. Accordingly, the two-step procedure was demonstrated for the late-stage labelling of probenecid, providing the desired compound **25** in 57% yield for the carboxylation step (see Supplementary Information Section 5.2).

Radioactive labelling is essential for bioactive molecule development and regulatory approval, but its costs can be extensive. To minimise this, the developed method relies on the controlled ex situ release of only 1.5 equivalents of $^{14}CO_2$ and can be easily set up on the benchtop. Moreover, the direct application of Ba$^{14}CO_3$, the primary carbon-14 reactant, in a late-stage carboxylation reaction avoids additional radioactive steps. The possibility of labelling a compound with carbon-14 was demonstrated in good yields (cmpd **2**). Next, the late-stage synthesis of a radiolabelled retinoic acid receptor alpha (RARα) agonist, tamibarotene (**30**), was demonstrated in 63% yield with only one step of manipulating the radioactive material. Finally, the synthesis of the olaparib precursor **31** was achieved in a satisfactory 67% yield. For the successful application of radiolabelled compounds in the drug development process, distinct molar activities ($A_m$) can be required. The use of low $A_m$ compounds (230 GBq mol$^{-1}$) can be limited to accelerator mass spectrometry (AMS), but with increased $A_m$ other important studies, such as animal mass balance, can be carried out[38]. With the described method it is possible to dilute the Ba$^{14}CO_3$ with Ba$^{12}CO_3$ to tune the carbon-14 incorporation, and thus obtain the desired $A_m$, such as for compounds **2** and **30**. In addition, if only Ba$^{14}CO_3$ is used for the gas release, full carbon-14 incorporation is obtained leading to a high-$A_m$ compound as demonstrated for compound **31**.

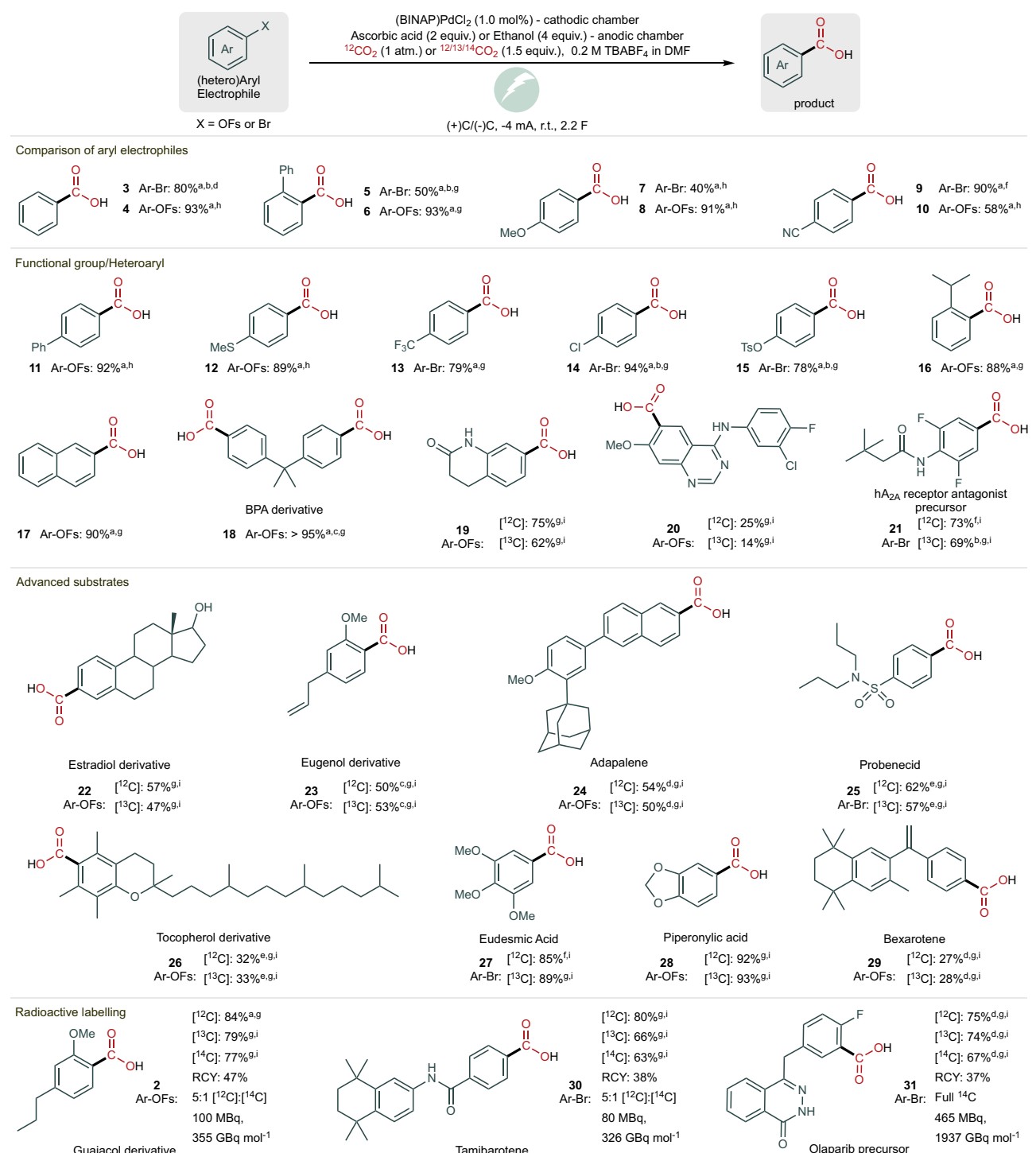

**Fig. 3 | Scope of aryl electrophiles.** All reactions were performed using 0.4 mmol of aryl electrophile. All reported yields are an average of at least two runs. RCY is used as an acronym for radiochemical yield. [a]Reaction performed with 1 atm of $^{12}CO_2$. [b]Reaction performed with −2 mA of constant current. [c]Reaction performed with −1 mA of constant current. [d]Reaction performed with 5 mol% of pre-catalyst. [e]Reaction performed with a constant potential. [f]Reaction performed using ascorbic acid as an additive. [g]Reaction performed using ethanol as an additive. [h]Reaction performed once with ascorbic acid as an additive and once using ethanol as an additive. [i]Reaction performed with the ex situ release of 1.5 equiv. of $CO_2$.

## Electrocarboxylation mechanistic study

After demonstrating the tolerability of the reaction conditions with a wide variety of motifs and applicability to the direct synthesis of carbon-labelled bioactive molecules, efforts were focused on elucidating the reaction mechanism. The reported catalytic cycle consists of an oxidative addition of $L_nPd(0)$ **I** to the aryl electrophile, leading to the formation of Pd(II) complex **III** (Fig. 4a)[16,36,41]. This complex has

been proposed to follow three different routes. Martin and coworkers proposed that the organometallic complex **III** can perform $CO_2$ migratory insertion[16], while Iwasawa coworkers reported that the reduction of **III** to **V** occurs prior to the $CO_2$ insertion[36]. Amatore and Jutand described that the triphenylphosphine ligated **III** undergoes an overall two-electron reduction to **VI** before the carboxylation step, which is suggested to occur outside the palladium coordination

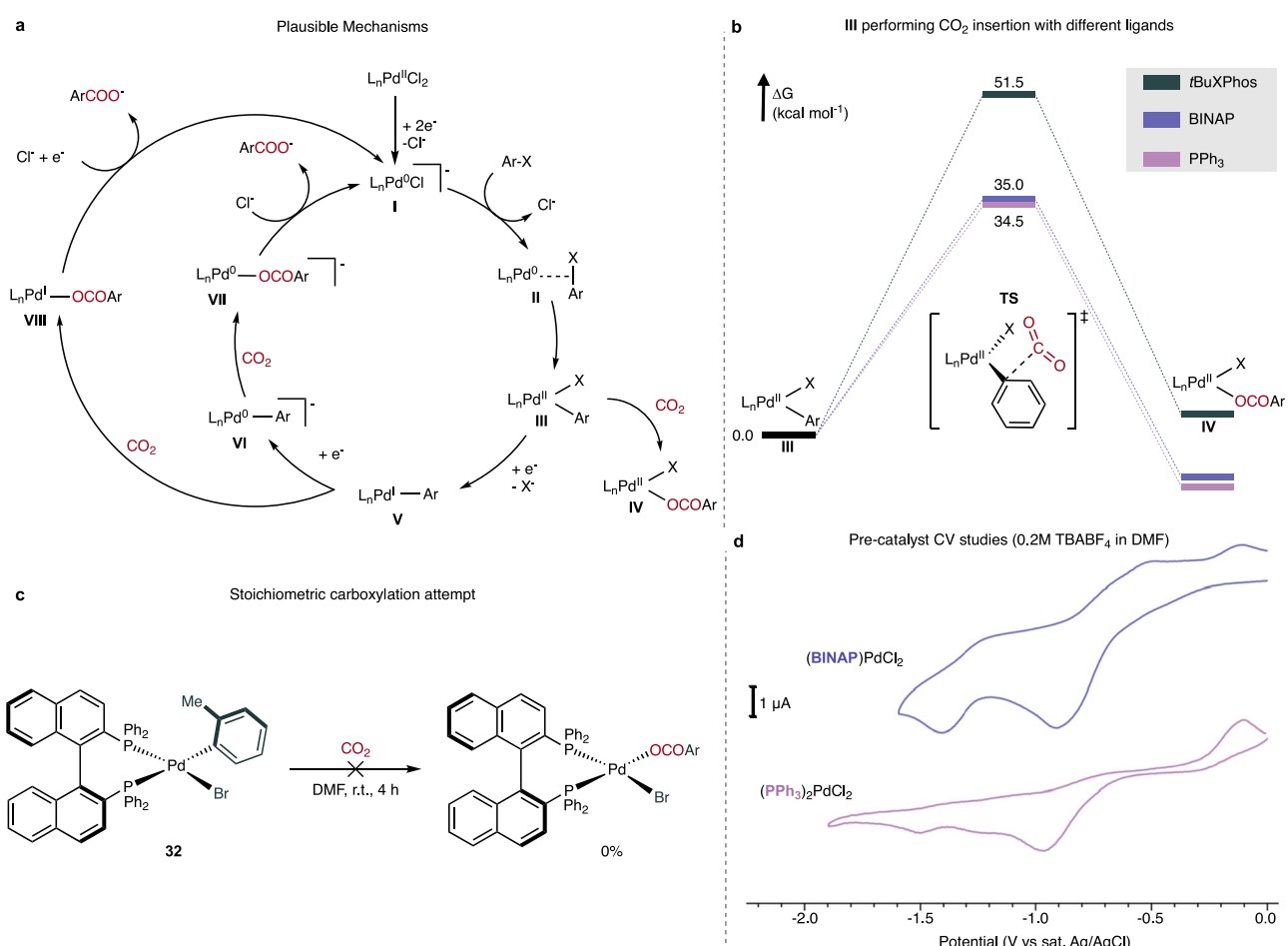

**Fig. 4 | Mechanistic study. a** Plausible catalytic cycles (based on reported studies[16,36,41], and experimental and computational analyses performed here). **b** DFT study for carboxylation of **III** with different ligands. **c** Reaction of Pd(II) complex **32** with $CO_2$. **d** Cyclic voltammetry of two different palladium pre-catalysts.

sphere[41]. We employed density functional theory (DFT) at the B3LYP-D3[IEFPCM] level to compute the barriers for $CO_2$ α-migratory insertion into complex **III** with different ligands (Fig. 4b). With 2-di-tert-butylphosphino−2′,4′,6′-triisopropylbiphenyl (tBuXPhos), $PPh_3$ and BINAP, the computed energy barriers range from 34 to 51 kcal/mol, which are incompatible with a room temperature reaction.

To further validate these results experimentally, we attempted the reaction of aryl-Pd(II) complex **32** with $CO_2$ at room temperature in DMF (Fig. 4c). After 4 h, no carboxylation product was observed, confirming that the $CO_2$ migratory insertion in complex **III** cannot occur under these reaction conditions. The combined results indicate that the active carboxylation species is not a Pd(II) complex. The palladium's ability to access odd oxidation states, thus allowing the existence of Pd(I)-species **V**, depends on the nature of the ligand. DuBois and coworkers demonstrated that bidentate ligands with larger bite angles can stabilise the Pd(I) complex[42]. This is based on electrochemical studies that revealed that $[Pd(dppm)_2](BF_4)_2$ undergoes an overall two-electron reduction, whereas 1,2-bis((diphenylphosphino)methyl)benzene (dppx) or 1,2-bis(diphenylphosphino)ethane (dppe) can undergo one-electron reduction[42]. A cyclic voltammetry (CV) study with the $(BINAP)PdCl_2$ pre-catalyst revealed that the two one-electron reduction peaks are present at −0.85 and −1.35 V vs Ag/AgCl (Fig. 4d and see SI Section 4.2), which was also observed by Amatore and Jutand for dppe[41]. In contrast, $(PPh_3)_2PdCl_2$, the pre-catalyst employed by Amatore and Jutand for the same transformation, displays one peak corresponding to a two-electron reduction process (Fig. 4d). In addition to the possibility of accessing a Pd(I) complex, higher stability of the catalyst due to the chelate effect is expected with

BINAP. This should inhibit catalyst deactivation due to palladium(0) cluster formation at the operating negative potentials.

Next, we computed the barriers for the carboxylation of aryl-Pd complexes **V** and **VI**, which showed that both have lower activation barriers than species **III**, consistent with a $CO_2$ migratory insertion step at room temperature (Fig. 5a). To confirm that a low-valent palladium complex is involved in the $CO_2$ migratory insertion step, we conducted CV studies using the aryl-Pd(II) complex **32**. Interestingly, in the presence of $CO_2$, a peak shift was observed at the reduction potential of the Pd(I)/Pd(0) wave (−1.56 V in argon to −1.68 V in $CO_2$) (Fig. 5b). Furthermore, two oxidation peaks (−0.95 V and −0.36 V) relative to complex **32** in argon are not present in $CO_2$ atmosphere (Fig. 5b). These results suggest an interaction between the low-valent palladium complex and $CO_2$. To determine if these observations with well-defined species can be translated to the catalytic reaction, we analysed the charge within the course of the reaction for p-methoxy aryl fluorosulfonate using constant potential electrolysis (CPE) (Fig. 5c). Such an analysis can provide insights into the kinetic profile of an electrochemical reaction, as under a CPE regime, the consumed charge is correlated with the reduction of an analyte, which for the described reaction is complex **III**. Although the concentration of **III** decreases with time, a linear decrease is observed, demonstrating that the reaction is of the order of 0 for the palladium catalyst and $CO_2$ (Fig. 5c and see SI Section 4.5). Consistent with this result is the additional observation that the reaction yield and conversion remain constant with double catalyst loading within the same reaction time with a −2 V (vs. Ag/AgCl) applied potential, indicating that the pre-catalyst concentration does not affect the reaction rate (Fig. 5d). The same trend

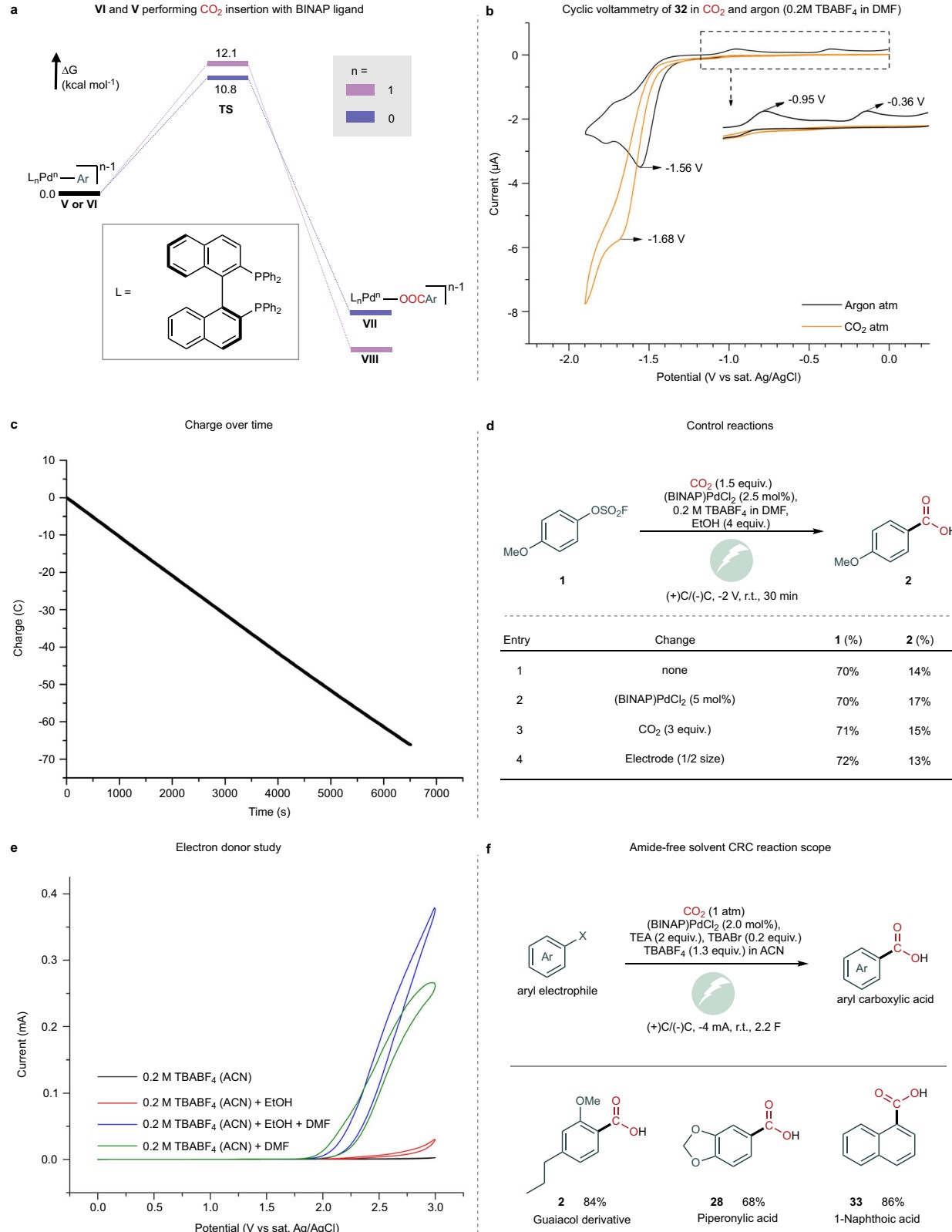

**Fig. 5 | Mechanistic study. a** DFT study of the carboxylation of **V** and **VI** with the BINAP ligand. **b** Cyclic voltammetry of oxidative addition complex **32** in argon and $CO_2$ atmosphere. **c** Consumed charge during the reaction. **d** Control reactions. **e** Effect of adding ethanol in the cyclic voltammetry in DMF and acetonitrile (ACN). **f** Scope of the carboxylation reaction with amide-free solvent.

was observed for increased $CO_2$ concentrations, implying that although carboxylation occurs within the palladium coordination sphere, it is not rate-determining.

## Anodic reaction investigation

With a more complete understanding of the carboxylation (cathodic) reaction, we turned our attention to the anodic compartment to elucidate the role of ethanol as an additive. Because of the wider redox window of acetonitrile, it was selected as the solvent of choice for a comparative CV study on the oxidation of DMF and ethanol (Fig. 5e). This indicates that DMF oxidation is more favourable than ethanol, which leads to the conclusion that the oxidation of DMF is feasible under our reaction conditions in a manner similar to that proposed by Ross et al. to generate a carbonium ion species[43]. Although this oxidation is feasible, trapping the carbonium ion species with a nucleophile leads to an *N*-alkoxymethyl-*N*-methylformamide product, which may explain the positive effect of added ethanol on the carboxylation reaction[44]. Thus, the requirement of DMF as a solvent was elucidated: it is the sacrificial reductant. This, in turn, makes the catalytic system a suitable candidate for the use of a non-amide-based solvent. This change is highly desirable and has only been previously reported by Hazari and co-workers using a nickel catalyst. Although sophisticated, it relies on DMAP-OED as an organic reductant, which is sensitive and expensive[26]. Thus, we decided to investigate whether the change of DMF to acetonitrile was feasible. While DMF oxidation occurs under the reported reaction conditions, with acetonitrile, the oxidation is not productive because of the high oxidation potential of the solvent. The use of ascorbic acid as an electron donor was also problematic due to poor solubility. We then envisioned that triethylamine could act as an electron donor, similarly as commonly employed in photoredox catalysis[45–47]. After a small optimisation, it was observed that bromide ions are required for solvent exchange to be viable. This was easily solved by adding TBABr (0.2 equivalents) to the commonly employed $TBABF_4$ (Fig. 5f). Thus, under slightly modified conditions, the reaction tolerated the change to a non-amide solvent and allowed the access of aryl carboxylic acids in good yields. Although the use of amide-based solvents for the small-scale synthesis of labelled bioactive compounds does not present an issue, the exchange for a safer solvent and the use of a commercial catalyst in a simple setup lay the foundation for the implementation of catalytic reductive carboxylations in the pharmaceutical industry[48].

In summary, the development of an efficient Pd-catalysed electrocarboxylation method for the late-stage carbon isotope labelling of aryl carboxylic acids is described. This general method can tolerate the use of only 1.5 equivalents of $^{14}CO_2$, making full radioactive labelling of carboxylic acids attainable in only one radioactive step. In the process, the reaction mechanism was studied, and a key low-valent aryl-Pd complex was proposed to undergo $CO_2$ migratory insertion in the cathodic chamber, while the DMF participated as the electron donor in the anodic chamber. With a clear understanding of both cathodic and anodic reactions, the use of acetonitrile as an amide-free solvent was feasible. Altogether, we anticipate that our general and mild late-stage carbon label incorporation method will expedite the use of $CO_2$ as a $C_1$ synthon for labelling strategies. Moreover, we expect that using a safer solvent with a commercial catalyst in a simple electrochemical setup will promote the implementation of CRC in the pharmaceutical industry.

## Data availability

The details about the materials and methods, experimental procedures, mechanistic studies, characterisation data, and NMR spectra generated in this study is available in the Supplementary Information. The Cartesian coordinates of all optimised structures are available from Supplementary Data 1. The authors declare that all other data supporting the findings of this study are available within the paper and its Supplementary Information files. Additional data is available from the corresponding author by request.

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

## Acknowledgements

The authors appreciate the financial support from the Danish National Research Foundation (grant no. DNRF118), NordForsk (grant no. 85378), the Independent Research Fund Denmark/Technology and Production Sciences, and Aarhus University. Support from the European Union's Horizon 2020 research and innovation programme under grant agreement no. 862179 and Marie Sklodowska-Curie grant agreement no. 859910 is also gratefully acknowledged. This publication reflects the views only of the authors, and the Commission cannot be held responsible for any use which may be made of the information contained therein. The authors also appreciate the financial support from The Novo Nordisk Foundation $CO_2$ Research Center (CORC). The authors are also highly thankful to CSCAA for the computing hours for the DFT study. K.H.H. additionally thanks the Research Council of Norway (no. 300769) and Sigma2 (no. nn9330k). C.S.D additionally thanks the Novo Nordisk Foundation under the grant agreement NNF23OC0081745.

## Author contributions

The project was designed by G.M.F.B. and T.S. The experimental data was obtained in collaboration by G.M.F.B., R.E., C.S.D., A.R-H., and K.T.N. Radioactive labelling was evaluated by G.M.F.B. with guidance from C.E. and J.B. G.M.F.B. conducted density functional theory studies with guidance from K.H.H. All authors contributed to the revision of the manuscript.

## Competing interests

T.S. is co-owner of SyTracks A/S, which commercialises COware®. The remaining authors declare no competing interests.
