## [Peer Review File · Nature Communications]

Efficient palladium-catalyzed electrocarboxylation enables late-stage carbon isotope labellingReviewers' Comments:

Reviewer #1:

Remarks to the Author:

This paper describes Pd-catalyzed electrochemical carboxylation of aryl bromides and fluorosulfonates, which has enabled efficient late-stage carbon isotope labelling by using only 1 mol% of Pd(BINAP)Cl₂ as catalyst. The most important finding of this paper is the high efficiency, which enables stoichiometric use of Ba¹⁴CO₃ as a source of ¹⁴CO₂ for this reaction. Mechanistic studies are also carried out to realize use of CH₃CN as solvent of the present reaction. However, most of the fundamental aspects of the results described in this paper have already been reported in previous papers by Fauvarque, Jutand, and Torii for Pd-catalyzed electrochemical carboxylation reactions and by Martin, Iwasawa, and Jana for Pd-catalyzed carboxylation reactions in combination with photoredox catalysts, although some optimizations have been carried out in the present method to realize low catalyst loading and a stoichiometric amount of CO₂.

Concerning the efficient capture of CO₂, such high efficiency has been noted in several cases, where a catalytic amount of CO₂ generated from formate salt is captured efficiently and introduction of ¹³C is also described there (Org. Lett., 2015, 17, 1814; Green Chem., 2023, 25, 6194). Comparison with the previous protocols by the present authors showed that reasonable results are obtained in some cases as shown in Figure 2d.

The reactive species for carboxylation has been discussed in the previous papers both electrochemically (Ref 41) and photochemically (Ref 36) and there seem to be no obvious new points in this paper (See also, Chin. Chem. Lett. 2021, 32, 1403). It is noted that clarification of the role of DMF as an electron-donor leads to the finding that acetonitrile can be employed for this reaction using Et₃N as an electron-donor, which expands the utility of the reaction.

All the experimental works are carried out soundly with sufficient care and the results support the claims by the authors without problem. There seems to be no problem in Supporting Information. Evaluation of this paper depends mainly on the utility of the present reaction as a method for the preparation of carbon-14 labeled aryl carboxylic acids, because the basic aspect of the present reaction is mostly dependent on the previously reported methods. My honest opinion is that although preparation of labeled carboxylic acids would be a very useful contribution to related research areas, the method itself is mostly dependent on the known methods and originality of the reaction is not sufficient. Considering all these things together, I do not think this paper is sufficiently novel to be accepted for Nature communications.

Reviewer #2:

Remarks to the Author:

In this manuscript, Rosas-Hernández, Skrydstrup and co-workers report a palladium-catalyzed electro carboxylation for an application to carbon isotope labeling. Despite the fact that the carboxylation of aryl halides has already been widely reported, limitations are remaining for some applications to isotopic carbon labelling, where the use of limited stoichiometry of CO₂ is necessary.

For this reason, the authors decided to develop a palladium catalyzed electro catalyzed carboxylation using stoichiometric amount of CO₂. After optimization, the authors managed to carboxylated aryl bromides and aryl fluorosulfates, allowing the carboxylation of substrates that are inaccessible by previous methods. Then, a broad scope of substrates has been carboxylate with global quite acceptable yields. An important effort was made to synthesize compounds with bioactive moieties or drug derivatives. Importantly, the methodology proved suitable to C¹³ and radiocarbon C¹⁴. To conclude the article, the authors realized an extensive mechanistic study (CV, DFT ...) in order to get a better understanding of the reactions taking place at the cathode and the anode. Particularly interesting is the effort to determine the catalytically active palladium species that should be operating during the key CO₂ insertion step. With a better understanding of the reaction mechanism, the authors managed to change the solvent from DMF to acetonitrile, which is an amide-free solvent more likely to be used in the pharmaceutical industry than DMF.

The authors have carried out a large amount of work and for the most part, the supporting information document is well written and clear. The work represents sufficient innovation for publication in Nature communication, if a number of modifications are brought to the manuscript.

1. In general, in the manuscript it would be of great help, if the authors could mention to which part (Fig, Scheme, page..) of the supporting information they are referring to.
2. One of the weakness of this article is the clarity of its optimization in the supporting information and in the manuscript:
 - a. The use of additives in the anodic chamber is not clearly mentioned. We do understand that a brief explanation is given at the end of the manuscript, but the authors should explain more the impact of those additives on the yield of the reaction. This is particularly important for general readership who might not be familiar with electrochemistry
 - b. For lower yield reactions, what is observed in the crude reaction mixture (unreacted starting material, byproducts ...)?
 - c. Please, be homogeneous choose either ascorbic acid or vitamin C for all the manuscript.
 - d. The authors mention that the reaction was first pre-optimized in a photoredox setup. Why? It will be nice to have this optimization disclosed in the supporting information.
 - e. The yields that are in Figure 2 are not fitting with the yields provided in the SI. Please, correct and clarify all this part.
 - f. How the authors can explain the fact that ascorbic acid is leading to no conversion with 1.5 equivalents of CO₂?
3. How could you explain the low yield of the substrate 20? Do you observe by products or just low conversion ?
4. Does the reaction work directly on heterocycles (for example bromopyridine)?
5. No example is disclosed with a EWG or EDG in meta position: is that tolerated or do you observe an impact on the yield?
6. The radiolabeling part is very interesting and the authors made a great effort for the labelling of pharmaceutically relevant moieties. It will be great to provide the RCY on figure 3. Maybe a comment could be added to the manuscript on the molar activities obtained: are they suitable for applications?
7. Page 6, the authors state that aryl fluorosulfates were chosen for their easy access from phenols. I do not fully agree with that. As reported in the SI, those reagents need to be synthesized in a two chamber apparatus and, in addition, required the in situ generation of HF. I would suggest modifying the sentence in the text.
8. Concerning the SI, I strongly suggest to properly highlighting in the SI the potential high danger of generating HF in the reaction that allows preparation of aryl fluorosulfates. It is mentioned, but only "Caution remarks »(p S41). It should properly be mentioned in the general procedure, and highlighted in bold.
9. Figure 3:
 - from product 11 to 31, it is not clearly mention which substrate was used ? OFs of Br? Please specify it clearly
 - On top of the Fig, in the reaction scheme (over the arrow), it is specified 1.5 equiv of CO₂. If I understand correctly, from product 3 to 18, 1 atm of CO₂ was used. Please, find a way to include both stoichiometry clearly in the scheme.
10. Products 14 and 15 show that, in a competitive substrate, OFs reacts selectively vs OTs and Cl. Nonetheless, it would be highly informative to know whether Ar-OTs, I, and Br work as substrates or not. Could the authors include that information in the revised manuscript?
In addition, aliphatic substrates are suitable?
11. In the supporting information: figure 3 we can see that the membrane is a bit dark. How the authors make sure that no more palladium is remaining on this membrane for the next reaction? Is that problematic?
12. It seems that there is an upper connection between cathode and anode compartments: have you ever tried to perform the reaction with CO₂ only in the cathode compartment?

Reviewer #3:

Remarks to the Author:

The manuscript by Skrydstrup and coworkers presents the development of an efficient palladium catalyst for electrochemical cross-electrophile coupling with stoichiometric CO₂. The scope was studied, also towards isotope labeling. The mechanism was studied by experiment and computation. The optimized catalyst proved broadly applicable. While other related e-CEC with CO₂ had previously developed, also with non-precious transition metals, but it appears that the new palladium catalyst is most efficient with equimolar amounts of CO₂. This is very important for applications to isotope labeling.

Given the topical interest in electrochemical couplings with ambient CO₂, I recommend publication of this fine manuscript after minor revision.

- 1) Cobalt catalysts have been employed for e-CEC with CO₂ (for example *Angew. Chem. Int. Ed.* 2020, 59, 12842): How do typical cobalt catalyst perform here?
- 2) Which other phenol derivatives can be used as the electrophile? Can tosylates or mesylates be used efficiently?
- 3) Specify carbon paper electrode.

We sincerely thank the referees and the editor for the in-depth feedback that they provided. We address all their questions via the following point-to-point responses:

Reviewer #1

Remarks to the Author:

This paper describes Pd-catalyzed electrochemical carboxylation of aryl bromides and fluorosulfonates, which has enabled efficient late-stage carbon isotope labelling by using only 1 mol% of Pd(BINAP)Cl₂ as catalyst. The most important finding of this paper is the high efficiency, which enables stoichiometric use of Ba¹⁴CO₃ as a source of ¹⁴CO₂ for this reaction. Mechanistic studies are also carried out to realize use of CH₃CN as solvent of the present reaction.

1) However, most of the fundamental aspects of the results described in this paper have already been reported in previous papers by Fauvarque, Jutand, and Torii for Pd-catalyzed electrochemical carboxylation reactions and by Martin, Iwasawa, and Jana for Pd-catalyzed carboxylation reactions in combination with photoredox catalysts, although some optimizations have been carried out in the present method to realize low catalyst loading and a stoichiometric amount of CO₂.

Answer: We acknowledge that our methodology builds upon the prior research of Fauvarque, Jutand, Torii, Martin, Iwasawa, and Jana, all of whom were cited in the introduction to establish the current state-of-the-art in the field. However, this study marks the initial instance of an effective near-stoichiometric carboxylation reaction of aryl electrophiles. Of utmost significance is the compatibility of our developed method with carbon-14, enabling the facile late-stage and single-step labeling of pharmaceutical precursors, a crucial aspect in drug development.

2) Concerning the efficient capture of CO₂, such high efficiency has been noted in several cases, where a catalytic amount of CO₂ generated from formate salt is captured efficiently and introduction of ¹³C is also described there (Org. Lett., 2015, 17, 1814; Green Chem., 2023, 25, 6194). Comparison with the previous protocols by the present authors showed that reasonable results are obtained in some cases as shown in Figure 2d.

Answer: The efficiencies of the research works mentioned by the reviewer, in which formate salts are used as CO₂ sources for carboxylation reactions, cannot be directly compared with those obtained with our methodology. The reported works are constrained by the use of activated substrates such as allenes (Org. Lett., 2015, 17, 1814) and benzylic fluorides (Green Chem., 2023, 25, 6194). The carboxylation of these types of substrates is favored by the generation of resonance-stabilized anions upon the reduction step. Our developed methodology works efficiently with aryl bromides and aryl fluorosulfates with high CO₂-capture efficiency, allowing us to run the carboxylation reactions in near-stoichiometric conditions.

Attempts to translate previous carboxylation reports by Martin, Iwasawa, Jana, Tsuji, Yu, and König demonstrate that the change from high CO₂ concentrations to only 1.5 equivalents is challenging. Reports with the use of nickel catalysis led to no carboxylic acid product, while the use of palladium led to considerably lower yields of **2** (Figure A1). The procedure reported by Iwasawa performed the best leading to **2** in a 68% yield, which is a value considerably lower than the one obtained using our methodology (85%). It is important to note this previous work requires 2.5 mol% palladium catalyst loading in conjunction with 2.5 mol% of an iridium-based photocatalyst and was set up inside a glovebox due to the use of the air-sensitive tBuXPhos ligand. This is in sharp contrast with our methodology which only requires 1 mol% of a commercially available pre-catalyst and can be setup on a benchtop. Besides the higher catalyst loading, the glovebox requirement is a major drawback when considering radiolabelling applications in the pharmaceutical industry. Additionally, we demonstrated the compatibility of our electrocatalytic system beyond the simple product **2**, including complex substrates with biological activity, highlighting its ability to perform late-stage radioactive labelling reactions on drug substrates.

Figure A1: Performing published CRC methods with 1.5 equivalents of CO₂ in the two-chamber (see SI section 3).^{16,17,19,23,25,27} (Figure 2 d in the main manuscript) ^a Yields were obtained by ¹H NMR spectroscopy of a crude reaction mixture with an internal standard.

3) The reactive species for carboxylation has been discussed in the previous papers both electrochemically (Ref 41) and photochemically (Ref 36) and there seem to be no obvious new points in this paper (See also, Chin. Chem. Lett. 2021, 32, 1403). It is noted that clarification of the role of DMF as an electron-donor leads to the finding that acetonitrile can be employed for this reaction using Et₃N as an electron-donor, which expands the utility of the reaction.

Answer: Jutand (J. Am. Chem. Soc. 1992, 114, 7076-7085) and Dubois (Inorg. Chem. 1991, 30, 417-427) previously reported that the reactivity of the palladium catalyst is highly dependent on the chemical structure of the ligand used. They demonstrated how the ligand's structure also governs the possibility of the Pd center accessing odd oxidation states, such as Pd(I). Within Jutand's study (J. Am. Chem. Soc. 1992, 114, 7076-7085) it was reported how changes in the ligand system can significantly alter the catalytic carboxylation reaction, with ligands such as DPPE not behaving as a catalyst, although able to carry out the carboxylation reaction stoichiometrically between the DPPE-aryl-palladium(II) complex and CO₂. Furthermore, Jutand demonstrated that using triphenylphosphine as the ligand, an overall 2-electron reduction process occurs, leading to a (PPh₃)₂-aryl-Pd(0) anion, which is in equilibrium with the free anion that can undergo carboxylation. With this mechanistic study, it was demonstrated that the Pd-catalyst is responsible for facilitating the reduction of the aryl halide and that it is not involved in the carboxylation step. This is in contrast with what has been proposed by Iwasawa and Yu who considered the possibility of accessing Pd(I) intermediates and that Pd intermediates are involved in the carboxylation steps.

This previous research demonstrates that the efficiency of the carboxylation cycle heavily depends on the type of supporting ligand. We, therefore, decided to explore the use of BINAP, an unexplored ligand in this transformation, and perform in-depth mechanistic studies. We demonstrated that, in contrast to Jutand's report, carboxylation occurs within a low-valent aryl-palladium complex. Furthermore, a study to understand the role of the solvent was also performed. As described by Martin (Chem 2021, 7, 2927), the utilization of amide-based solvents for carboxylation reactions is a bottleneck hindering CRC reactions from industrial applications. Thus, this study also aimed at further understanding the necessity, or not, for DMF to be present to facilitate catalytic carboxylation, which led to the possibility of exchanging the reaction solvent for ACN. The use of DMF for large-scale production of pharmaceuticals is often avoided, and thus this mechanistic study added to the understanding of how catalytic carboxylation reactions can occur in alternative solvents.

4) All the experimental works are carried out soundly with sufficient care and the results support the claims by the authors without problem. There seems to be no problem in Supporting Information. Evaluation of this paper depends mainly on the utility of the present reaction as a method for the preparation of carbon-14 labeled aryl carboxylic acids, because the basic aspect of the present reaction is mostly dependent on the previously reported methods. My honest opinion is that although preparation of labeled carboxylic acids would be a very useful contribution to related research areas, the method itself is mostly dependent on the known methods and originality of the reaction is not sufficient. Considering all these things together, I do not think this paper is sufficiently novel to be accepted for Nature communications.

Answer: The obtention of carbon-labeled compounds is of utmost importance for pharmaceutical and agrochemical development. The efficient incorporation without the need for excess labeling material assures the production of less radioactive waste, which for carbon-14 comes with very expensive handling and storage due to a half-life of 5730 years. Methods for the obtention of radiolabeled carbonyl compounds using CO as the carbon source have been described in depth, but the utilization of radiolabeled CO₂ under catalytic conditions is still in its infancy, though Ba¹⁴CO₃ is the first carbon-14 source. This lack of labelling protocols is due to CO₂'s thermodynamic and kinetic inertness, thus presenting difficulties in its activation, even more so considering a diluted source.

Reviewer #2

Remarks to the Author:

In this manuscript, Rosas-Hernández, Skrydstrup and co-workers report a palladium-catalyzed electro carboxylation for an application to carbon isotope labeling. Despite the fact that the carboxylation of aryl halides has already been widely reported, limitations are remaining for some applications to isotopic carbon labelling, where the use of limited stoichiometry of CO₂ is necessary.

For this reason, the authors decided to develop a palladium catalyzed electro catalyzed carboxylation using stoichiometric amounts of CO₂. After optimization, the authors managed to carboxylate aryl bromides and aryl fluorosulfates, allowing the carboxylation of substrates that are inaccessible by previous methods. Then, a broad scope of substrates has been carboxylate with global quite acceptable yields. An important effort was made to synthesize compounds with bioactive moieties or drug derivatives. Importantly, the methodology proved suitable to C13 and radiocarbon C14. To conclude the article, the authors realized an extensive mechanistic study (CV, DFT ...) in order to get a better understanding of the reactions taking place at the cathode and the anode. Particularly interesting is the effort to determine the catalytically active palladium species that should be operating during the key CO₂ insertion step. With a better understanding of the reaction mechanism, the authors managed to change the solvent from DMF to acetonitrile, which is an amide-free solvent more likely to be used in the pharmaceutical industry than DMF.

The authors have carried out a large amount of work and the for the most part, the supporting information document is well written and clear. The work represents sufficient innovation for publication in Nature communication, if a number of modifications are brought to the manuscript.

1) In general, in the manuscript it would be of great help, if the authors could mention to which part (Fig, Scheme, page..) of the supporting information they are referring to.

Answer: As suggested by the reviewer, we have indicated across the main text of the manuscript to which part of the Supporting Information we are referring.

2) One of the weakness of this article is the clarity of its optimization in the supporting information and in the manuscript:

a. The use of additives in the anodic chamber is not clearly mentioned. We do understand that a brief explanation is given at the end of the manuscript, but the authors should explain more the impact of those additives on the yield of the reaction. This is particularly important for general readership who might not be familiar with electrochemistry

Answer: The necessity of additives in the anodic chamber is now discussed in the *Electrocarboxylation optimisation* subsection of the *Results* section:

“Using an additive, including ascorbic acid, ethanol, or triethylamine, in the anodic chamber was essential for obtaining good yields of the carboxylic acid (see SI section 2). In the presence of the additive, the potential at the cathode is not limited by the rate of the oxidation reaction since such additives should display a lower oxidation potential compared to the solvent, thus obtaining the product with higher yields.”

Additionally, a more detailed discussion was added in the supporting information (section 2):

“Although the use of ascorbic acid represents a renewable feedstock in comparison to triethylamine, we decided to further investigate if other additives can be used in the anodic chamber. In the described electrochemical setup, the potential in the counter electrode will increase until a compound in the anodic chamber is oxidized. Thus, either an additive or the solvent must be oxidized to provide the electrons for the reaction of interest in the cathodic chamber. It is of utmost importance that the oxidation rate does not limit the potential at the working electrode during the constant-current electrolyzes in the two-electrode set-up. If the necessary potential to drive the oxidation reaction at the required current is too high, the potential at the working electrode might not be negative enough to reduce the key palladium intermediary with appreciable rates, leading to low yields and undesired side products. The absence of an additive led to lower reaction yields (37%), demonstrating that the direct oxidation of the solvent under the reaction conditions requires a higher oxidation potential, whereas the addition of ethanol was shown to facilitate this process, obtaining yields of 87%.”

b. For lower yield reactions, what is observed in the crude reaction mixture (unreacted starting material, byproducts ...)?

Answer: For the reactions in which the use of electron-rich aryl bromides as substrates led to lower yields, unreacted substrate was observed in the crude reaction mixture. For reactions with electron-poor aryl fluorosulfates, competitive direct reduction occurs, ultimately leading to phenols.

The rationalization for this is described in the main text:

“Electron-rich or hindered aryl bromide electrophiles give lower yields due to the more demanding oxidative addition, whereas electron-rich or hindered aryl fluorosulfates provide the desired product in good yields.”

“An electron-withdrawing motif decreases the reduction potential of aryl fluorosulfate, which can lead to substrate reduction as a side reaction.”

c. Please, be homogeneous choose either ascorbic acid or vitamin C for all the manuscript.

Answer: We have now modified the main text and supporting information files, and now ascorbic acid has been used everywhere.

d. The authors mention that the reaction was first pre-optimized in a photoredox setup. Why? It will be nice to have this optimization disclosed in the supporting information.

Answer: Due to the availability of photochemical equipment in our laboratory, a ligand optimization was carried out using photoredox catalysts to reduce the palladium catalyst and enable turnover.

A general procedure and discussion for this has been added to the supporting information Section 2.1.:

“Inside the glovebox, a pressure tube glassware (Figure A2) was charged with base, additives, photocatalyst, palladium precursor, and ligand. To that, 0.5 mL of solvent was added, followed by the addition of the sacrificial reductant (by volume with a micropipette) and the substrate (by weighting directly inside the glassware), and another 0.5 mL of solvent was added, rinsing the walls of the glassware. The glassware was closed and taken outside the glovebox, where it was cooled down with ice and connected to a vacuum line (Figure A2) that is connected to a CO₂ cylinder. The atmosphere was changed 3 times with the application of the vacuum until the solution started bubbling and leaving under a CO₂ atmosphere for 1 minute. Then, it was displaced at the top of a stirring plate at a 5 cm distance to a Kessil LED A160WE Tuna Blue 40 W and allowed to stir overnight.

Figure A2: Glassware used for the photoredox reactions. (Figure 1 in the SI)

Initially, a ligand screening was performed under photoredox conditions, (Scheme 1). In these experiments, BINAP, along with tBuXPhos and BrettPhos gave the best results, with BINAP presenting the highest yield and selectivity towards the desired product. Since BINAP is a less electron-rich phosphine compared to the other two ligands, the oxidative addition complex should have a lower reduction potential, thus allowing it to be applied for electron-poor substrates; the described advantages of BINAP in comparison to tBuXPhos and BrettPhos led us to continue with BINAP as the ligand of choice. Under photoredox conditions, Iwasawa has shown that a higher proton concentration in the medium leads to greater amounts of the side product **S1**. This can be partially avoided with the use of bases, but even with 3 equivalents of cesium carbonate, 37% of product **S1** can still be observed. In an electrochemical setup, it is possible to separate the reduction and oxidation reactions and thus control the proton concentration near the palladium catalyst, which led us to transpose the BINAP-palladium catalytic system to an electrochemical setup.”

Scheme A1: Ligand screening. (Scheme 1 in the SI) ^a NMR yield calculated with 1,3,5-trimethoxybenzene as internal standard”

e. The yields that are in Figure 2 are not fitting with the yields provided in the SI. Please, correct and clarify all this part.

Answer: The authors thank the reviewer for the precision. This was modified in the manuscript.

f. How the authors can explain the fact that ascorbic acid is leading to no conversion with 1.5 equivalents of CO₂?

Answer: With the reaction optimization at 1 atmosphere of CO₂ completed, we wanted to ensure that the system was still selective under low CO₂ concentrations. For that, near-stoichiometric CO₂ release from barium carbonate

was envisioned, due to the adaptability for carbon-12, carbon-13, and carbon-14 presented by this carbonate source. Initially, trials adding BaCO₃ directly to the anodic chamber were explored (Scheme 6 entries 1 and 2) due to the formation of the protons in the oxidation reaction. This was a failed attempt and can be attributed to the very slow/lack of CO₂ release using this strategy. To guarantee a fast CO₂ release, a third chamber was added to the system (Scheme 6 entries 3-6), where the gas release occurs, ensuring a successful reaction. Thus, the lack of reactivity in Scheme 6 entry 2 is not due to the use of ascorbic acid, but due to the poor conditions for CO₂ release.

This was more clearly described with an additional text in the supporting information section 2.3.:

“With the optimized reaction conditions using 1 atmosphere of CO₂, we attempted to corroborate that the system was still very selective towards the carboxylic acid under low concentrations of CO₂ (Scheme 6). We first attempted to use the protons formed at the anodic chamber to react them directly with barium carbonate for CO₂ release. This was attempted with a simple divide cell as demonstrated in Scheme 6 (top right picture), with the addition of barium carbonate directly in the anodic chamber. This reaction failed, but since we still had the possibility of using ascorbic acid as the electron donor, we attempted to use it as both an electron donor for the reduction and a proton donor for the CO₂ release in the same glassware, but this was also a failed attempt. With this, we decided to use an additional third chamber that can be used specifically for the CO₂ release from barium carbonate and camphorsulfonic acid as demonstrated in Scheme 6 (bottom right picture). Analysis of the best solvent mixture for the gas release was also done, with layering ethylene glycol in water giving the best result (Scheme 6 entry 6).”

3) How could you explain the low yield of the substrate 20? Do you observe by products or just low conversion?

Answer: Due to the electron-poor nature of substrate 20, a competitive reduction reaction toward the phenol product is favored. The fact that competitive reduction of electron-poor aryl fluorosulfates can occur is discussed in the manuscript: “An electron-withdrawing motif decreases the reduction potential of aryl fluorosulfate, which can lead to substrate reduction as a side reaction.”

4) Does the reaction work directly on heterocycles (for example bromopyridine)?

Answer: The use of heterocycles has been demonstrated in the substrate scope (Figure A3). The direct carboxylation of a substituted quinazoline was demonstrated to provide carboxylic acid **20**. Additionally, substituted 2-quinolone, chromane, and methylenedioxybenzene were also tolerated and provided products **19**, **26**, and **28** respectively. Likewise, substrate **31** contains a phthalazone heterocycle and could be obtained with the reported procedure.

Figure A3: Heterocycles examples from the scope (Figure 3 in the main manuscript). All reactions were performed using 0.4 mmol of aryl electrophile. All reported yields are an average of at least two runs. RCY is used as an acronym for radiochemical

yield. ^a Reaction performed with 1 atm of ¹²CO₂. ^b Reaction performed with -2 mA of constant current. ^c Reaction performed with -1 mA of constant current. ^d Reaction performed with 5 mol% of pre-catalyst. ^e Reaction performed with a constant potential. ^f Reaction performed using ascorbic acid as an additive. ^g Reaction performed using ethanol as an additive. ^h Reaction performed once with ascorbic acid as an additive and once using ethanol as an additive. ⁱ Reaction performed with the ex-situ release of 1.5 equiv. of ¹²CO₂.

5) No example is disclosed with a EWG or EDG in meta position: is that tolerated or do you observe an impact on the yield?

Answer: The use of EDG in the meta position was tolerated for a variety of substrates (Figure A4), being demonstrated in up to 92% yield for the synthesis of aryl carboxylic acids **19**, **22**, **26**, **27**, **28**, and **31**. Additionally, we have also demonstrated that EWG can be tolerated at the meta position with the synthesis of carboxylic acid **21** in up to 73% yield.

Figure A4: Meta-substituted examples from the scope (Figure 3 in the main manuscript). All reactions were performed using 0.4 mmol of aryl electrophile. All reported yields are an average of at least two runs. RCY is used as an acronym for radiochemical yield. ^a Reaction performed with 1 atm of ¹²CO₂. ^b Reaction performed with -2 mA of constant current. ^c Reaction performed with -1 mA of constant current. ^d Reaction performed with 5 mol% of pre-catalyst. ^e Reaction performed with a constant potential. ^f Reaction performed using ascorbic acid as an additive. ^g Reaction performed using ethanol as an additive. ^h Reaction performed once with ascorbic acid as an additive and once using ethanol as an additive. ⁱ Reaction performed with the ex-situ release of 1.5 equiv. of ¹²CO₂.

6) The radiolabeling part is very interesting and the authors made a great effort for the labelling of pharmaceutically relevant moieties. It will be great to provide the RCY on figure 3. Maybe a comment could be added to the manuscript on the molar activities obtained: are they suitable for applications?

Answer: The RCY was added to Figure 3 (as shown in Figure A5):

Figure A5: Radio-labelled examples from the scope (Figure 3 in the main manuscript). All reactions were performed using 0.4 mmol of aryl electrophile. All reported yields are an average of at least two runs. RCY is used as an acronym for radiochemical yield. ^a Reaction performed with 1 atm of ¹²CO₂. ^b Reaction performed with -2 mA of constant current. ^c Reaction performed with -1 mA of constant current. ^d Reaction performed with 5 mol% of pre-catalyst. ^e Reaction performed with a constant potential. ^f

Reaction performed using ascorbic acid as an additive. ^g Reaction performed using ethanol as an additive. ^h Reaction performed once with ascorbic acid as an additive and once using ethanol as an additive. ⁱ Reaction performed with the ex-situ release of 1.5 equiv. of ¹²CO₂.

Additionally, a comment on the molar activities has been added to the manuscript:

“For the successful application of radiolabelled compounds in the drug development process, distinct molar activities (Am) can be required. The use of low Am compounds (230 GBq mol⁻¹) can be limited to accelerator mass spectrometry (AMS), but with increased Am other important studies, such as animal mass balance, can be carried out.³⁸ With the described method it is possible to dilute the Ba¹⁴CO₃ with Ba¹²CO₃ to tune the carbon-14 incorporation, and thus obtain the desired Am, such as for compounds **2** and **30**. Additionally, if only Ba¹⁴CO₃ is used for the gas release, full carbon-14 incorporation is obtained leading to a high-Am compound as demonstrated for compound **31**.”

7) Page 6, the authors state that aryl fluorosulfates were chosen for their easy access from phenols. I do not fully agree with that. As reported in the SI, those reagents need to be synthesized in a two chamber apparatus and, in addition, required the in situ generation of HF. I would suggest modifying the sentence in the text.

Answer: Our approach is not necessary for the synthesis of aryl fluorosulfates, this is just the method of choice in our laboratory due to the wide availability of two-chamber glassware. Roth and Fuller (J. Org. Chem. 1991, 56, 3493-3496) have demonstrated how fluorosulfonic anhydride can be used for the obtention of the same product. Additionally, several reports demonstrate how this reaction can be carried with the direct use of sulfuryl fluoride gas, such as by Sanford (J. Am. Chem. Soc. 2017, 139, 4, 1452–1455), Shen (Org. Lett. 2023, 25, 13, 2318–2322), and Qin (Chem. Asian J. 2017, 12, 2323 –2331).

This comment suggesting alternative methods for the synthesis of aryl fluorosulfates has been added to the SI section 5.1.

8) Concerning the SI, I strongly suggest to properly highlighting in the SI the potential high danger of generating HF in the reaction that allows preparation of aryl fluorosulfates. It is mentioned, but only “Caution remarks »(p S41). It should properly be mentioned in the general procedure, and highlighted in bold.

Answer: A comment has been added to the general procedure section 5.1 where the fluorosulfation reaction is described, the comment was highlighted in bold and with the pictograms for HF.

9) Figure 3:

- from product 11 to 31, it is not clearly mention which substrate was used ? OFs of Br? Please specify it clearly.

- On top of the Fig, in the reaction scheme (over the arrow), it is specified 1.5 equiv of CO2. If I understand correctly, from product 3 to 18, 1 atm of CO2 was used. Please, find a way to include both stoichiometry clearly in the scheme.

Answer: In light of the referee’s comment, we have corrected Figure 3 indicating which substrate was used to perform all the carboxylation reactions. We have also modified the figure to include both CO₂ reaction conditions.

10) Products 14 and 15 show that, in a competitive substrate, OFs reacts selectively vs OTs and Cl. Nonetheless, it would be highly informative to know whether Ar-OTs, I, and Br work as substrates or not. Could the authors include that information in the revised manuscript? In addition, aliphatic substrates are suitable?

Answer: In relation with question 9, we think there was confusion. In reality (can be seen in SI section 6.6 – and now is more clear in figure 3), it was a competitive reaction of a Br vs OTs or Cl. It has been previously shown that

aryl fluorosulfates react faster when in a competitive reaction with aryl bromides (Angew. Chem. Int. Ed. 2018, 57, 6858–6862). This indicates that in the case of aryl fluorosulfates the same or higher chemoselectivity is expected.

The use of different aryl electrophiles was attempted (Figure A1). Aryl tosylates are unreactive in the reaction conditions, this is correlated with the more demanding oxidative addition compared to aryl fluorosulfates, as previously described by Albanese-Walker (Org. Lett. 2009, 11, 7, 1463–1466). Aryl iodides yield a complex mixture of products, this can be associated with the lower reduction potential of aryl iodides, thus leading to competitive substrate reduction, which is unproductive. Attempts to increase the catalyst loading to favor the carboxylation pathway were not attempted. Aryl chlorides are also another class of more challenging electrophiles, and no carboxylic acid product could be observed for substrate **A3**.

Figure A1: Attempt to perform carbonylation of different aryl electrophiles.

Alkyl halides are a more challenging class of electrophiles when compared with aryl halides, as described by Beller (Angew. Chem. Int. Ed. 2005, 44, 674–688). When considering that the catalyst is unable to activate aryl tosylates, we expect that aliphatic halides are not compatible with the described conditions.

11) In the supporting information: figure 3 we can see that the membrane is a bit dark. How the authors make sure that no more palladium is remaining on this membrane for the next reaction? Is that problematic?

Answer: After the completion of the electrochemical reaction, a cleaning procedure with acetone, water, and aqua regia is performed. Such cleaning protocol ensures the removal of any residues of palladium. This information has now been added at the beginning of Section 6 of the Supporting Information.

12) It seems that there is an upper connection between cathode and anode compartments: have you ever tried to perform the reaction with CO₂ only in the cathode compartment?

Answer: We did not attempt to carry out the electrochemical carboxylation reactions excluding CO₂ from the anodic compartment. Both compartments are connected in the H-cell to equilibrate the pressure inside the H-cell and prevent solvent exchange between chambers. We anticipate that excluding CO₂ from the anodic chamber would not have any effect on the performance of the reaction since the molecule that is oxidized is the added additive or the solvent.

A comment about the need for a gas bridge has been added in the SI (Page 10):

“It is important to state that both glasswares have an upper gas bridge in between the anodic and cathodic chambers to avoid different overhead pressures inside the H-cell and prevent solvent exchange between chambers. For the case with the near stoichiometric CO₂ release, the cathodic chamber was chosen to be the one closer to the gas release chamber, as demonstrated in Figure 4 and 5.”

Reviewer #3

Remarks to the Author:

The manuscript by Skrydstrup and coworkers presents the development of an efficient palladium catalyst for electrochemical cross-electrophile coupling with stoichiometric CO₂. The scope was studied, also towards isotope labeling. The mechanism was studied by experiment and computation. The optimized catalyst proved broadly applicable. While other related e-CEC with CO₂ had previously developed, also with non-precious transition metals, but it appears that the new palladium catalyst is most efficient with equimolar amounts of CO₂. This is very important for applications to isotope labeling. Given the topical interest in electrochemical couplings with ambient CO₂, I recommend publication of this fine manuscript after minor revision.

1) Cobalt catalysts have been employed for e-CEC with CO₂ (for example *Angew. Chem. Int. Ed.* 2020, 59, 12842): How do typical cobalt catalyst perform here?

Answer: Two catalytic systems were attempted with the developed setup and substrate 1:

The cited manuscript (*Angew. Chem. Int. Ed.* 2020, 59, 12842) utilizes a mixture of cobalt acetate (10 mol%) and triphenylphosphine (20 mol%) and does not provide the desired carboxylic acid **2**. In the crude reaction mixture analysis by ¹H NMR and GCMS, it is only possible to observe starting material, phenol, and trace amounts of the hydrodehalogenation side product.

Additionally, Wang and co-workers (*Chem. Commun.*, 2020, 56, 14416-14419) have reported a cobalt-catalyzed carboxylation of aryl bromides and chlorides using cobalt bromide (10 mol%) and neocuproine (10 mol%). The use of this catalytic system in the described reaction conditions was also attempted and did not lead to the desired carboxylic acid **2**. In the crude reaction mixture analysis by ¹H NMR and GCMS, it is only possible to observe the starting material, and hydrodehalogenation side product. This demonstrates that although the catalytic system can perform oxidative addition into the aryl fluorosulfate the catalyst is not selective at low CO₂ concentrations.

A detailed experimental procedure of these two attempts was added to section 3 of the supporting information.

2) Which other phenol derivatives can be used as the electrophile? Can tosylates or mesylates be used efficiently?

Answer: The reaction with an aryl tosylate did not give products, that can be associated with the more demanding oxidative addition compared to aryl fluorosulfates, as described by Albanese-Walker (*Org. Lett.* 2009, 11, 7, 1463-1466). Although the use of mesylates was not attempted, they have been reported to react similarly to tosylates, indicating that with the reported conditions mesylates are unreactive.

3) Specify carbon paper electrode.

Answer: The type of carbon paper is specified both in the general methods and in the general procedure in the SI:
“Toray carbon paper 090 wet-proofed.”

Reviewers' Comments:

Reviewer #2:

Remarks to the Author:

The authors have properly answered to the comments highlighted by this reviewer.
I support publication of the article and congratulate the authors for their work.

Reviewer #3:

Remarks to the Author:

The revised manuscript has addressed all of my previous comments in a suitable manner.